cognition/behaviour

problem solving, *Larus delawarensis*, aquatic bird, cognition, animal behaviour, means-end understanding

**Author for correspondence:**
Jessika Lamarre
e-mail: jlamarre@mun.ca

# Waterbird solves the string-pull test

Jessika Lamarre[1] and David R. Wilson[2]

[1]Cognitive and Behavioural Ecology Program, and [2]Department of Psychology, Memorial University of Newfoundland, St John's, Canada

JL, 0000-0002-7688-2068; DRW, 0000-0002-6558-6415

String-pulling is among the most widespread cognitive tasks used to test problem-solving skills in mammals and birds. The task requires animals to comprehend that pulling on a non-valuable string moves an otherwise inaccessible food reward to within their reach. Although at least 90 avian species have been administered the string-pull test, all but five of them were perching birds (passeriformes) or parrots (psittaciformes). Waterbirds (Aequorlitornithes) are poorly represented in the cognitive literature, yet are known to engage in complex foraging behaviours. In this study, we tested whether free-living ring-billed gulls (*Larus delawarensis*), a species known for their behavioural flexibility and foraging innovativeness, could solve a horizontal string-pull test. Here, we show that 25% (26/104) of the ring-billed gulls that attempted to solve the test at least once over a maximum of three trials were successful, and that 21% of them (22/104) succeeded during their first attempt. Ring-billed gulls are thus the first waterbird known to solve a horizontal single-string-rewarded string-pull test. Since innovation rate and problem-solving are associated with species' ability to endure environmental alterations, we suggest that testing the problem-solving skills of other species facing environmental challenges will inform us of their vulnerability in a rapidly changing world.

## 1. Introduction

Cognition is challenging to assess in wild animals because testing paradigms often require individuals living under similar conditions to be tested repeatedly [1,2]. Administering cognitive tests where wild animals rear their offspring can overcome this challenge because breeders often return reliably to a known location. Researchers can therefore make access to a nest, den or burrow part of a cognitive task [3,4], or can introduce a foraging test within the animal's defended breeding territory [5,6]. The most common cognitive test requires individuals to overcome a novel obstacle blocking access to food; their success at solving these foraging puzzles indicates their problem-solving skills and innovation potential [7–10]. One of the most extensively studied

and most-implemented foraging puzzles for mammals and birds is the string-pull test, where food is visible to the animal but accessible only by pulling on a string attached to the reward (review by [11]). Conditions of the string-pull test paradigm can be made more complex by using multiple string choices, such as presenting paired baited and unbaited strings or using crossed strings [11]. In order to succeed at these more complicated designs, many argue that animals must display insight and means-end understanding to comprehend that pulling on a string with no inherent value has the positive effect of retrieving the otherwise inaccessible food [12–15]. While learning through trial-and-error can contribute to solving string-pull tests, some animals solve the test spontaneously on their first attempt [16–18].

Since the first written record of string-pulling in European goldfinches (*Carduelis carduelis*) dating back more than 2000 years [11], string-pull tests have been used as a measure of cognition in at least 68 mammalian species and 90 avian species, though all but five of the avian species have been perching birds (order Passeriformes; $N = 49$ species) or parrots (order Psittaciformes; $N = 36$ species) [11,15,19]. The glaucous-winged gull (*Larus glaucescens*; order Charadriiformes) is the only waterbird that has been tested with a string-pull test, and it was considered unsuccessful after a small sample of two captive individuals failed to retrieve the food (personal observation of T.A. Obozova, as reported in [11]). In fact, waterbirds are poorly represented in the cognitive literature generally, perhaps because they were never expected to be as intelligent as corvids and parrots given their smaller relative brain size [20–22]. We are aware of only two studies that have experimentally tested problem-solving skills in waterbirds. Castano *et al.* [23] reported that none of the 53 free-living Olrog's gulls (*Larus atlanticus*) that showed interest in a transparent box containing food could open the lid to retrieve the food. Danel *et al.* [24] tested 26 free-living brown skuas (*Catharacta antarctica* ssp. *lonnberg*) with an exclusion test in which each subject was presented with two opaque cups—one empty and one containing food. Subjects correctly chose the baited cup over the empty cup 74% of the time if they had been shown the contents of both cups beforehand, and 64% of the time if they had been shown the empty cup only [24]. Despite the paucity of cognitive testing, waterbirds are suspected to display behaviours often associated with advanced cognition, including possible tool-use in southern black-backed gulls (*Larus dominicanus*) [25], and bait-fishing in herring gulls (*Larus argentatus*) [26] and several species of heron and egret (family Ardeidae) [27–31]. Gulls and skuas (order Charadriiformes) are also well known for their ability to exploit novel environments [32–34], which usually requires advanced cognitive abilities [35–39].

Here, we show that ring-billed gulls (*Larus delawarensis*) can solve a configuration of the string-pull test. Our study therefore expands the list of taxa that can solve this puzzle to include waterbirds (clade Aequorlitornithes), which encompasses the shorebirds, gulls and auks (Charadriiformes), the flamingos and grebes (Mirandornithes), the sunbittern, kagu (a terrestrial species), and tropicbirds (Phaethontimorphae), and the loons, pelicans, herons, petrels, penguins, storks, frigatebirds, sulids, cormorants, shoebill and hamerkop (Aequornithes) [40]. Like many waterbirds, ring-billed gulls exhibit several characteristics associated with advanced cognition, including that they are long-lived foraging generalists [41] with high behavioural flexibility in their foraging ecology and an ability to thrive in novel environments [33,41,42].

# 2. Methods

## 2.1. Study sites and subjects

We studied ring-billed gulls at four breeding colonies in Newfoundland, Canada (figure 1). The colonies ranged from urban to remote and thus had access to the diverse foods and foraging opportunities that are characteristic of this species. While remote-nesters have typically conserved their historical diet of fish, aquatic and terrestrial invertebrates, and the occasional bird or rodent [43–45], urban-nesters forage almost exclusively on anthropogenic food sources, including refuse, grains and agricultural waste [33,42,46].

Adult gulls were tested during their late-incubation period in June 2020, when adults are easiest to capture and less likely to abandon their nests [47–49]. We monitored the colonies' laying dates to determine when their last week of incubation would occur, assuming that incubation lasted for 26 days after clutch completion [41]. The colonies' breeding periods were each staggered by approximately 1 week, which permitted us to study them in succession (Long Pond, 7−14 June; Spaniard's Bay, 17−21 June; Old Perlican, 22−26 June; Salmonier, 27−30 June).

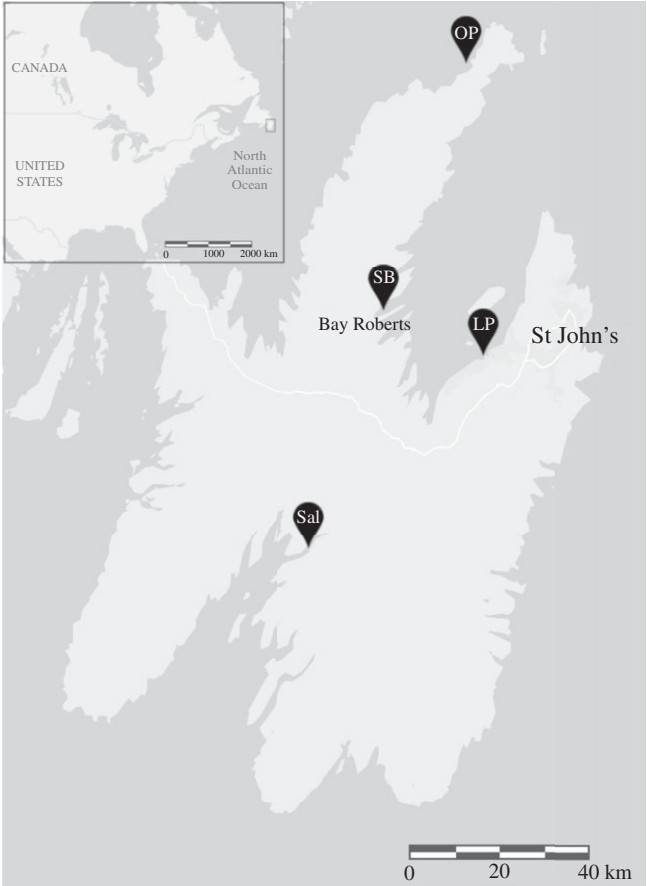

**Figure 1.** Locations of the four colonies studied in Newfoundland, Canada. The Long Pond (LP; 47°31′09.8″ N, 52°58′33.6″ W) and Spaniard's Bay (SB; 47°35′51.8″ N 53°16′48.7″ W) colonies are situated in urban environments, whereas the Old Perlican (OP; 48°05′15.7″ N 53°01′20.6″ W) and Salmonier (Sal; 47°08′11.0″ N 53°28′48.6″ W) colonies are situated in more remote areas. All colonies are connected to the mainland by a sandbar, except for Old Perlican which is an island situated 600 m from the mainland. All colonies are bordered by the Atlantic Ocean.

We haphazardly targeted incubating gulls and captured them from the nest using a hand net or noose trap over 3 days at the Long Pond colony, and over 2 days at the Spaniard's Bay, Old Perlican and Salmonier colonies. Our goal was to capture and band one or both adults from at least 40 nests per colony, though this was only possible at Long Pond ($N = 46$ individuals from 43 nests) and Spaniard's Bay ($N = 40$ individuals from 40 nests), which were the largest colonies (estimated to be greater than 300 pairs each at the time of the study). The Old Perlican and Salmonier colonies were smaller (less than 150 pairs each at the time of the study), and we could only capture 22 and 25 individuals (each from a different nest), respectively. In general, the gulls learned quickly to avoid us, which made it difficult and increasingly disruptive to capture additional individuals or the mates of those already captured. Although we targeted gulls with unhatched eggs at capture, some nests hatched during the following days of testing. In all cases, all testing was complete before the chicks became mobile and ventured out of their nest cup (less than 7 days old [50,51]).

Captured adults were banded with a permanent and uniquely numbered Canadian Wildlife Service band on their left leg and a plastic colour band (green, blue, pink, purple or yellow) on their right leg. The colour band ensured that the gulls recorded on video during subsequent cognitive tests belonged to the specific nest being tested and that they could be distinguished from their mate. During banding, we collected morphological measures and a blood sample from each bird as part of our longitudinal research on these birds. All methods were performed under appropriate permits (Canadian Wildlife Service Scientific Permit, number SC4049; Environment and Climate Change Canada Scientific Permit to Capture and Band Migratory Birds, numbers 10 890 and 10 890B) and were approved by the Memorial University Animal Care Committee (number 19-03-DW).

To contain chicks that might eventually hatch, to prevent potential pilfering from neighbours during the string-pull tests and to minimize opportunities for social learning, we constructed a fence around the

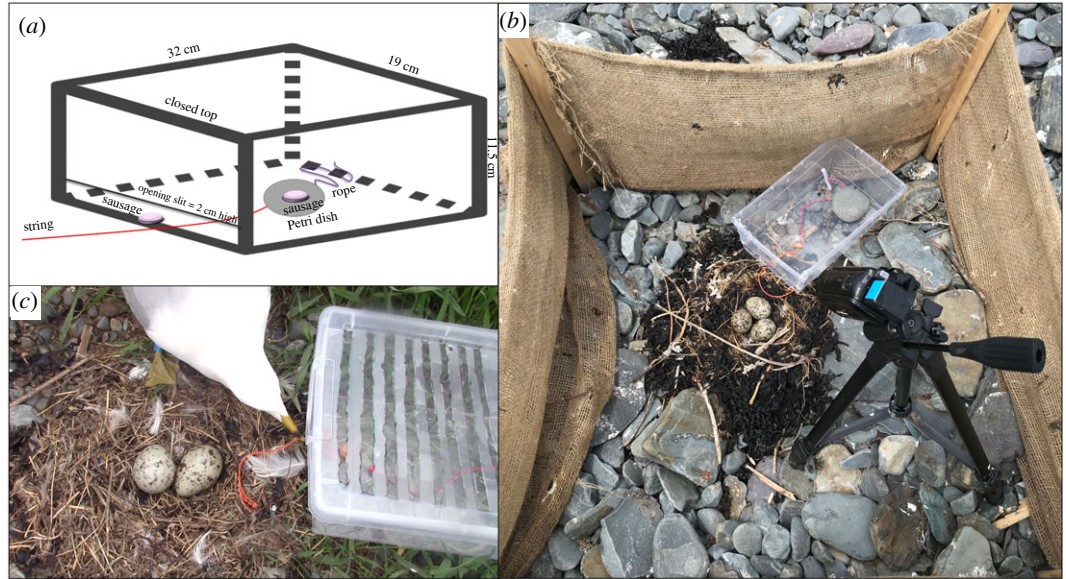

**Figure 2.** Horizontal version of the string-pull test used here to assess cognition in ring-billed gulls. (*a*) Schematic of the string-pull test with the lid on; the only way to access the sausage inside the box is to pull horizontally on the string that extends from a plastic Petri dish inside the box through the open slit at the base of the front panel. The Petri dish is tied to the back of the box by a long rope which does not prevent the dish from exiting the box through the open slit but prevents the gull from flying away with the Petri dish. (*b*) Photograph of the fenced nest during the last habituation trial in which the sausage on the Petri dish is accessible through the lidless top or by pulling on the string through the open slit at the base of the front panel. The box was pegged to the ground with a rock placed inside it to prevent it from moving. (*c*) Photograph of a string-pull test trial in which a banded gull (blue colour band) is pulling on the string before successfully solving the test. Because the lid of the box was slightly frosted, slits of 1 cm width were made to provide of a better view of the box's contents without providing access through the top. For the last habituation trial and for all three string-pull test trials, a video camera was placed inside the fence to determine whether the gull present during the trial was the banded parent or its unbanded mate.

nest of each captured adult, as in our previous study [52]. Four wooden posts (2.5 × 5.1 × 122 cm) were inserted partially into the ground in a square arrangement (1.3 × 1.3 m) centred on the nest. Semi-transparent synthetic burlap (90 cm height) was wrapped around the posts, stapled to them and fastened to the ground with rocks along with the bottom periphery (figure 2). At construction, the burlap was rolled onto itself from the top to stand at a height of 15 cm above the ground; the low initial height of the fence reduced visual disturbance at the site and encouraged parents to return to their nests and resume incubation more quickly. The height of the burlap fence was increased gradually throughout the following day to a height of 50 cm before starting the cognitive tests. We then confirmed that the raised enclosure did not impair the parents' ability to fly in and out of their nest.

## 2.2. String-pull test

We used a horizontal version of the string-pull test [11,15] to investigate gulls' problem-solving skills. The original vertical version of the string-pull test was designed for perching species and presented them with food hanging on a string that must be pulled up by the subject using its feet and beak [7,12,53]. In comparison, the horizontal design is often used with mammalian species and non-perching birds [7,11,54]. Ring-billed gulls are non-perching birds with palmate feet, and it is unlikely that they could grasp a string with their feet to hold it in place between pulls, as is necessary in the vertical configuration. The testing apparatus comprised a transparent plastic box (32 × 19 × 11.5 cm) with a removable lid and a 2 cm-high open slit across the base of the front panel (figure 2). The task required a gull to pull on a string protruding through the open slit in order to retrieve a food item placed in a Petri dish from inside the box (figure 2). The Petri dish was placed 10 cm away from the open slit, far enough to be inaccessible without pulling on the string but close enough to keep the gulls motivated to access the food reward.

Once banding was completed at a given colony, we placed a lidless version of the string-pull box beside each fenced nest, oriented such that the open slit rested on the rim of the nest (figure 2). We

then began a series of habituation trials in which we attempted to teach the gulls to associate the string-pull box with food, as has been done by similar studies using test boxes [54,55]. During the first habituation trial, we placed one piece of hot dog sausage (5 g each) at the edge of each box's open slit where the bird could grab it directly with its bill, and then left the colony and gave the parents 30 min to return to their nests and consume the food. We repeated this procedure two more times that day, and two more times the following morning, for a total of five habituation trials per nest. The fifth habituation trial differed from the first four in that we only tested two nests at a time (order selected to minimize colony disturbance), for 15 min instead of 30 min. These changes allowed us to record the trial with a video camera placed on a tripod next to the nest (Canon VIXIA HF R800 video recorder; 1920 × 1080 resolution, 35 mbps using MP4 compression, 60 fps). Before commencing the fifth habituation trial, we also removed any sausage remaining from previous trials, and then placed one piece of sausage in a plastic Petri dish (4 cm diameter transparent Petri dish with 1 cm high walls) placed at the centre of the floor of the box. A red string tied to the Petri dish passed through the open slit at the base of the box's front panel and extended 10 cm beyond the box (figure 2). We placed a second piece of sausage at the edge of the box's open slit, beside the string, to encourage the gulls to investigate the string. During this final habituation trial, the gulls could access the interior sausage through the lidless top or by pulling on the string horizontally to slide the Petri dish out of the box. We provided access to the full string-pull test set-up during this last habituation trial to avoid presenting novel objects (the string and the Petri dish) during the first test trial. After each of the five habituation trials, we noted whether any sausages had been consumed, though, in the first four trials, we had no way of knowing which parent had eaten them.

After completing the fifth habituation trial, we removed all sausages from the nests and immediately commenced the first string-pull test (13.00–18.00). It was identical to the fifth habituation trial, except that the lids were fastened to the string-pull boxes so that the gull could only obtain the internal sausage by pulling on the string. Gulls were also given 10 min instead of 15 min, since most gulls returned to their nests within 1 or 2 min or not at all during the final 15 min habituation trial. After testing a pair of nests, we removed the Petri dishes and any remaining food before moving to the next pair of nests. The following day, we administered a second string-pull test in the morning (06.00–11.00) and a third test in the afternoon (11.00–16.00), such that every nest was tested three times. The protocol used during the second and third string-pull tests was identical to that used during the first test. Since we could not control for food drive, we continued to place an easily accessible piece of sausage at the open slit of the box for all testing trials as a way of determining whether the birds recognized it as food and were motivated to eat it. Their willingness to eat the slit sausage during a trial was used to infer that they also recognized the piece of sausage in the Petri dish as a high-value reward.

We stopped conducting habituation trials or string-pull tests at a given nest if all eggs or chicks had been depredated or disappeared. Once the third string-pull test was complete, we removed all string-pull test boxes and the fences surrounding the nests and moved to the next colony.

## 2.3. Video analysis

We analysed gulls' behaviours from the video recordings of the fifth habituation trial and the three string-pull test trials using BORIS event recording software (v. 7.9 RC1) [56]. First, we determined whether or not a gull returned to the nest during the habituation trial or the string-pull test trial and, if it did, whether the gull was the banded or unbanded parent. It was always possible to identify an unbanded gull as the parent given their inclination to incubate within seconds of returning to their nest. There were eight instances of pilfering by a neighbouring gull (three during the fifth habituation trial, five during the string-pull test trials), but the thieves were only successful at stealing the sausage at the rim of the box before a parent returned to the nest and chased them out. There were no instances of a gull solving the test while pilfering, probably because they were always chased out by the parent within seconds of landing in the nest. For banded gulls, we confirmed that the colour of their leg band matched our records for that nest. We recorded whether the attending parent ate the easily accessible sausage placed at the slit of the test apparatus. We considered that the gull was attempting to reach the food reward if it pecked the box beyond simply eating the easily accessible sausage, or inserted its bill into the slit and attempted to grab the Petri dish directly, as seen in our trial recordings (electronic supplementary material, movie S1). We considered that a gull successfully solved the test if it extracted the Petri dish by pulling on the string with its beak (the Petri dish was never extracted any other way) and then consumed the sausage.

There were nine instances where both parents were present at the nest during a test trial. In six of those cases, neither parent interacted with the box while their mate was present; most of the time, their overlap at the nest lasted only long enough to switch incubation duty. For the other three instances, a parent interacted with the box while its mate was present. There was one occurrence of a banded bird solving the test while the unbanded mate looked on (electronic supplementary material, movie S2). Since this occurred during the third string-pull test trial, there was never an occasion for the unbanded mate to use this learning experience in a subsequent trial. The other two occurrences where both mates were present at the nest also happened during the last string-pull test trial and none of the gulls solved the test despite all interacting with the box. As such, we assume that social learning was unlikely to have enhanced the birds' ability to solve the string-pull test.

## 3. Results

We administered string-pull tests at 124 intact nests. At 31 of these, neither parent returned to the nest during any of the string-pull tests. Among the remaining 93 nests, a total of 138 different individual parents (excluding thieves) returned to the nest during at least one string-pull test. Of the 93 nests and 138 parents that were exposed to at least one test, 104 individuals (75%) from 84 nests attempted to solve the string-pull test during at least one of the nest's three trials (table 1). In the remaining nine nests, the parent present during the string-pull test never attempted to solve it. During a typical string-pull test trial, a parent returned to its nest within 2 min of the researcher leaving and resumed incubation either before or after investigating the test box. If they investigated the box during a trial, they usually began by consuming the sausage placed at the slit beside the string. They then either ignored the box (i.e. did not attempt to reach the food reward) or interacted with it further by putting their bill inside the slit and attempting to grab the Petri dish directly, by pecking gently at the box or by pulling on the string. Gulls attempting, but failing, to solve the string-pull can be viewed in our trial recordings (electronic supplementary material, movie S1).

Gulls from all four colonies successfully solved the string-pull test (table 1; electronic supplementary material, movie S2). Of the 104 individuals that attempted to solve the string-pull test at least once, 25% (26/104) of them solved the test at least once by extracting the Petri dish and consuming the sausage (table 1; electronic supplementary material, movie S2). Twenty-one per cent (22/104) of them solved the test during their first exposure to it (table 1). Three gulls repeated their success during a subsequent exposure (43%, or three out of seven gulls that had at least one attempt following their first success), and four gulls that solved the test in an earlier trial then failed to repeat their success during a subsequent trial (47%, or four out of seven gulls that had at least one attempt following their first attempt). Because we could not control or predict which mate would return to the nest during a given trial, several individuals were presented with the string-pull test only once, whereas others were present for all three trials.

Although not reflected in the results above, we note that eight gulls also pulled on the string to extract the Petri dish containing the sausage during the fifth habituation trial (electronic supplementary material, movie S3), instead of taking the sausage directly through the lidless top. Since the majority of birds present during this last habituation trial obtained the food reward from the lidless top, we still consider that this is a familiarization trial rather than a test trial, despite this small number of birds obtaining the sausage by pulling on the string. Of these eight successful individuals, three of them were never present during the string-pull test trials (one at Long Pond, one at Old Perlican and one at Salmonier), four of them solved it again during a subsequent test trial (three at Long Pond, one at Old Perlican) and one of them never solved it again despite showing interest in the box (Long Pond). This last bird appears to have accidentally pulled on the string as it was grabbing the sausage at the slit of the box during the final habituation trial; it then kept pulling on the string afterwards to obtain the food reward (electronic supplementary material, movie S3, second clip).

## 4. Discussion

We provide the first evidence that a waterbird, the ring-billed gull, can solve the horizontal configuration, single-rewarded string condition of the string-pull test. Furthermore, we show that this result is repeatable across four different colonies of wild birds, despite obvious differences in their proximity to urban centres and thus in their foraging opportunities.

**Table 1.** Summary of the number of nests and ring-billed gulls studied, and the number that participated in tests and solved them, among the four colonies tested. Where fractions are presented, the numerator provides the value of the variable and the denominator denotes the relevant comparison group. For example, at the Long Pond colony, there were 13 nests where a parent solved at least one of the string-pull tests, out of 34 nests where a parent attempted to solve at least one of the string-pull tests (i.e. 38%).

| variable | Long Pond | Spaniard's Bay | Old Perlican | Salmonier | total |
|---|---|---|---|---|---|
| nests that were administered at least one string-pull test | 41 | 36 | 22 | 25 | 124 |
| nests where at least one parent was present for at least one string-pull test | 36 | 28 | 17 | 12 | 93 |
| nests where at least one parent attempted to solve the test at least once | 34 | 28 | 14 | 8 | 84 |
| nests where an attempt was successful | 13/34 (38%) | 7/28 (25%) | 3/14 (21%) | 1/8 (13%) | 24/84 (29%) |
| gulls that were present for at least one string-pull test (excluding thieves) | 59 | 42 | 23 | 14 | 138 |
| gulls that were present that attempted to solve the test at least once (excluding thieves) | 44/59 (75%) | 36/42 (86%) | 16/23 (70%) | 8/14 (57%) | 104/138 (75%) |
| gulls that attempted to solve the test and succeeded at least once | 14/44 (32%) | 7/36 (19%) | 4/16 (25%) | 1/8 (13%) | 26/104 (25%) |
| gulls that solved the test on their first attempt | 11/44 (25%) | 6/36 (17%) | 4/16 (25%) | 1/8 (13%) | 22/104 (21%) |
| gulls that solved the test on their second attempt | 2/11 (18%) | 1/11 (9%) | 0/3 (0%) | 0/3 (0%) | 3/28 (11%) |
| gulls that solved the test on their third attempt | 1/3 (33%) | 0/3 (0%) | 0/1 (0%) | 0/1 (0%) | 1/8 (13%) |

Previous studies have been criticized for concluding whether or not a species is successful at the string-pull test based on small samples of captive birds (see critiques by [11,19]). Indeed, we are aware of only two studies that tested the string-pull test performance of more than 10 wild birds. Heinrich [12] tested 50 common ravens (*Corvus corax*) in the wild with vertical string-pull tests that remained unsolved throughout trials lasting 3 days each. During a second experiment, seven of 27 (26%) wild-caught common ravens solved the test after being exposed to it in captivity for 0.5–9 h each [12]. Audet *et al*. [19] tested wild-caught birds in a captive setting only; they reported that 18 of 42 (43%) Barbados bullfinches (*Loxigilla barbadensis*) and two of 31 (6%) Carib grackles (*Quiscalus lugubris*) solved the string-pull test at least once over 10 trials lasting 5 min each. Only two bullfinches and one grackle solved the test on their first attempt, though all solvers remained successful during subsequent trials [19]. Given the high success rate of bullfinches (43%), Audet *et al*. [19] concluded that their findings were further evidence of the impressive cognitive abilities of a species that was already known for its foraging innovations [57]. Since our study limited the number of trials per subject to three, we argue that a success rate of 21% on the first attempt and 25% overall is strong

evidence that ring-billed gulls are proficient at solving this configuration of the string-pull test. We consider our conclusions robust and representative of the species because they are based on a large sample of 104 gulls (those that attempted to solve the test at least once) distributed among 84 nests and four colonies that encompass the diverse urban and rural foraging opportunities that ring-billed gulls naturally exploit [33,41]. Nonetheless, we note that success repeatability was low among the few gulls that were present for a trial subsequent to one where they solved the test. Indeed, three gulls were able to repeat their success while four failed to do so. If we include the results from the fifth habituation trial, these numbers increase to seven birds that successfully retrieved the food reward during a subsequent trial and five that failed to retrieve the food during a subsequent trial. Further research is thus needed to clarify whether the birds that failed to repeat their success happened to have pulled on the string without an understanding of the task, whereas the birds that did repeat their success did understand the task or at least learned how to solve it from a previous attempt. Administering more complex conditions of the string-pull paradigm is required to properly assess the mechanisms underlying wild gulls' probability of solving these cognitive tests.

Ring-billed gulls are thus the first waterbirds shown to be capable of solving a horizontal configuration of the string-pull test using a single-baited string. This finding indicates that gulls, and possibly other waterbirds, are well-suited to engage in cognitive tasks and are thus candidates for more advanced puzzles that are more informative about animals' cognitive abilities than a single iteration of the string-pull test can be. Cognition has seldom been studied in this avian group, despite at least 17 species being known to bait-fish [26–30,58] and two other species being observed using a tool for preening [25,59]. Indeed, we are aware of only four studies that have experimentally tested waterbird cognition. These include two studies that experimentally tested the problem-solving skills of waterbirds (Olrog's gull: [23] brown skua: [24]), though neither employed the string-pull test paradigm. A third study showed a small number of horned puffins (*Fratercula corniculate*, $n = 5$) to be unsuccessful at an object permanence test [60] which tests whether an individual comprehends that an object continues to exist when it is suddenly hidden from view [61,62]. A fourth study showed that glaucous-winged gulls are capable of social learning because they solved a foraging puzzle only by watching a trained conspecific solve it first (Obozova *et al.* [63]). The paucity of cognitive research on waterbirds is surprising because the life-histories and foraging strategies that many waterbirds exhibit are often associated with enhanced brain development [64,65]. In general, delayed maturity and long lifespans provide more time for brain development [66,67] and are associated with greater encephalization and cognitive abilities that help long-lived species adapt to changing environments over time [65,68]. The majority of waterbirds have a slow pace of life in which they remain sexually immature for several years and have lifespans spanning decades [22,69–71]. Sociality and lifelong monogamy are also generally associated with higher intelligence because they require animals to navigate complex social systems and maintain a long-term relationship with a mate [72–74]. Since many waterbirds, including gulls, are colonial and pair for life [69,71], we might expect them to require advanced cognition to establish their place among conspecifics, choose a cognitively suitable mate and successfully reproduce.

Brain size and cognition are associated with foraging ecology in birds. Lefebvre *et al.* [35] first showed that, across 17 avian orders, species with greater relative forebrain size demonstrate greater foraging flexibility and more feeding innovations. Furthermore, exploiting multiple food types at higher trophic levels (e.g. omnivore, carnivore) is linked to greater relative brain size [2,75,76], and generalist foraging strategies are associated with higher innovation rates and enhanced cognition [77,78]. Although foraging flexibility and feeding innovations are less common in waterbirds that have evolved morphological and physiological foraging specializations (e.g. plunge-divers and pursuit-divers; [57]), many species in this clade are higher order predators that demonstrate foraging innovativeness (e.g. families Laridae, Stercorariidae and Ardeidae), which suggests that considerable cognitive abilities may exist in this group [39,77]. Indeed, there is a large body of literature describing feeding innovations in opportunistic waterbirds in general and in gulls in particular. Gulls' foraging innovations tend to be based on the birds' ability to recognize and exploit a wide range of novel anthropogenic food sources, including refuse [33,46,79], commercial fisheries [80–82], mink farms [83,84] and agricultural fields [42,85,86]. Given ring-billed gulls' long lifespan, delayed maturity, sociality, lifelong monogamy, generalist feeding strategy and ability to exploit novel anthropogenic food sources [41], it is not surprising that many individuals in our sample were capable of solving the string-pull test.

Innovative birds are more likely than less innovative birds to subsist when faced with habitat destruction and reduced access to natural food sources [65,87,88]. Indeed, innovation rates and problem-solving skills positively correlate with a species' ability to colonize new areas [38,89,90], to

persist through urbanization [78,91,92] and to withstand extinction [93]. Given that human activities continue to drastically affect aquatic ecosystems, it is important that researchers and managers continue to study waterbird cognition, as it is a critical determinant in overcoming their ongoing challenges relating to diminished foraging opportunities and habitat destruction [94,95]. Experimentally testing the problem-solving skills of animals can be used as a proxy to determine their innovative potential [96] and thus their vulnerability to environmental challenges [94,95]. We suggest that various configurations and conditions of the string-pull test outlined here be administered to other wild waterbirds as a convenient tool for assessing innovation rates and identifying species that might be particularly vulnerable to environmental alteration. Assessing the string-pull test performance of other waterbirds will also help broaden our understanding of animal cognition beyond the heavily studied passerine and psittacine species. This is important in order to provide a framework within which to compare cognitive skills that might be better suited to different taxonomic groups based on the type of habitat that each exploits.

Ethics. All methods were performed under appropriate permits (Canadian Wildlife Service Scientific Permit, number SC4049; Environment and Climate Change Canada Scientific Permit to Capture and Band Migratory Birds, numbers 10 890 and 10 890B) and were approved by the Memorial University Animal Care Committee (number 19-03-DW).

Data accessibility. The dataset used in this study is available in the Dryad Digital Repository: https://doi.org/10.5061/dryad.d51c5b03z [97].

The data are provided in the electronic supplementary material [98].

Authors' contributions. Conceptualization was done by J.L.; experimental design was done by J.L. and D.R.W.; fieldwork was done by J.L. and D.R.W.; video coding was done by J.L.; resources were provided by D.R.W.; writing the original draft was done by J.L.; editing was done by J.L. and D.R.W.

Competing interests. At the time of consideration and publication of this manuscript, David R. Wilson was a member of the Royal Society Open Science Editorial Board but was involved in no way with the assessment of the manuscript.

Funding. Funding was provided by the Natural Sciences and Engineering Research Council of Canada (CGS-M to J.L. and a Discovery Grant to D.R.W. (RGPIN-2015-03769)).

Acknowledgements. We thank two anonymous reviewers for comments that improved an earlier draft of the manuscript.

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
