## [Peer Review File · Royal Society Open Science]

Review History

RSOS-211343.R0 (Original submission)

Review form: Reviewer 1

Is the manuscript scientifically sound in its present form?

Yes

Are the interpretations and conclusions justified by the results?

No

Is the language acceptable?

Yes

Do you have any ethical concerns with this paper?

No

Have you any concerns about statistical analyses in this paper?

No

Recommendation?

Major revision is needed (please make suggestions in comments)

Comments to the Author(s)

Review

Manuscript number: RSOS-211343

Title: Waterbird solves the cognitive string-pull test

The authors have written an interesting study about string-pulling performance in wild ring-billed gulls. What I particularly like is the field aspect of this study and the number of individuals tested (124 individuals, four breeding colonies), it sounds like it required significant work and I really appreciate the research effort. Although they are crucial, there are still few experiments assessing problem solving in seabirds, and wild populations more generally. It was also interesting to see that some gulls solved the task while performing incubation duties, which validates the use of such setting for further wild cognitive experiments with this species. There are some points regarding how the results are interpreted which may benefit from greater parsimony though (e.g., as it is, the string-pulling test presented here does not allow assessing means-end understanding). In my opinion, the authors are testing whether gulls can spontaneously pull on a single rewarded string. Perhaps the authors should introduce the study and discuss the results with this aspect in mind, and with more economy in the explanations. Suggestions are outlined below.

Title

Perhaps replace the term "waterbird" by a term more specific about the species in question (or family). Waterbirds vary significantly in their ecology, morphology, and behaviour (e.g. gulls are extremely different from flamingos, penguins, or pelicans, which are also Aequorlithornithes) I also disagree with "the cognitive" aspect of the string-pull test: pulling on a single rewarded string does not allow such claim

Abstract

Means-end understanding is involved when animals solve different experimental conditions, for instance when they show goal-directedness, rely on proximity, and/ or do not depend on visual feedback (see review of Jacobs and Osvath, 2015).

Introduction

Lines 29-31: I really like this

Lines 35-36: one of the most extensively studied

Lines 37-39: this is an overclaim, it depends on the experimental conditions administered, perhaps be more specific

Lines 41-43: this sentence is not necessary (see Jacobs and Osvath 2015)

Lines 58-59: be careful, tool use is not always cognitively complex, and the interpretations made on puffins have been highly reassessed (e.g. Farrar 2020 and the work that followed e.g. Sándor and Miklósi 2020)

Line 64: it is not yet a cognitive challenge, this is the first interesting step, but more conditions are necessary to be able to safely speak about "cognitive capacities", especially with the use of the horizontal configuration

Line 66: the kagu is not a waterbird

Line 69: perhaps add s at generalist

Methods

Lines 73-78: for the future, it would be interesting to get data on other variables such as sex or personality, in order to interpret whether and how these variables affect performance on the task

Line 80: replace least likely by less likely

Line 113: what do you mean by "privacy"? Other gulls are usually chased by the parents, perhaps you meant privacy from the other partner? If yes, it is the same as the risk for social learning

Lines 113-122: perhaps the fence is too close from the nest? Was there a specific reason for this setting? Perhaps in the future the fence could be one meter away

Also, in video S1 sometimes we have the impression that the gull is a bit disturbed by the box. I understand that it might be difficult to run problem-solving experiments in the field and that the authors wanted to test breeding individuals as it is more convenient, but perhaps in the future you should put the box a bit far away to allow the birds to be more at ease

Lines 140-141: and this horizontal configuration is also considered less cognitively difficult than the vertical one, but of course it is more appropriate for non-perching birds

Line 143: perhaps "configuration" instead of rendition?

Lines 150-154: it's just my opinion, but I would test each nest once at a time. If a problem occurs with the apparatus, for instance if a string detaches from the box accidentally, it is usually safer to focus observations on one nest

Lines 158-167: I appreciate the fact that authors first ensured to teach the gulls to associate the food with the string. Perhaps another learning phase might have been interesting to add after the habituation phase. Often animals are taught to pull on the string too, which represents a necessary step to proceed to further, more complex experimental string-pulling conditions (see the variety of conditions shown in the review made by Jacobs and Osvath 2015).

Lines 193-195: the gull did not attempt "to solve the task", perhaps reformulate by "the gull was attempting to reach the food reward if it..."

Lines 205-206: there could be low-level social learning mechanisms in play (for instance social motivation, social enhancement) because the birds interacted with the box, perhaps be more specific

Discussion

Lines 264-265: stay parsimonious, you did not relate your finding to other cognitive tasks or behavioural traits, and did not use various two-strings experimental conditions

Line 270: beware, this study tested a very limited number of captive individuals

Line 276: add the reference in parentheses

Lines 283-284: perhaps cognitive abilities may be selected also within the sexual pair

Line 300: perhaps a more cautious sentence is required, see my comment above

Lines 312-313: explain why it is important to study other bird species outside passerine and psittacine species (from the fundamental cognitive point of view, not for the conservation one)

Line 330: be more cautious

References

Use italics for latin names and recheck the writing of the references (e.g. 4)

Video S1: correct the legend sentence shown in the movie

Review form: Reviewer 2

Is the manuscript scientifically sound in its present form?

No

Are the interpretations and conclusions justified by the results?

No

Is the language acceptable?

Yes

Do you have any ethical concerns with this paper?

No

Have you any concerns about statistical analyses in this paper?

No

Recommendation?

Major revision is needed (please make suggestions in comments)

Comments to the Author(s)

The authors present a nice study on string-pulling in a previously untested species of waterbirds. The authors employed a widely used string-pulling paradigm to assess the problem-solving abilities of wild ring-billed gulls. Overall, this is an interesting work and I think the paper would make a valuable contribution to the field as it adds important data to body of literature on string-pulling that has been heavily dominated by work on Passeriformes and Psittaciformes. However, there are a few aspects of the paper that I would like to see clarified / improved.

My main concern is that in its current form, the manuscript creates an impression that successful performance in a single string task is still commonly considered to show means-end understanding, which is not the case. It is widely accepted that a successful pulling up a string only proves that a species possesses the required sensorimotor capacities and are therefore suitable for further testing as it is unclear what cognitive abilities underpin a successful response. The basic string-pulling skills do not really say much about cognitive skills without other string-pulling tasks testing for causal understanding, or connectivity skills. Thus I would suggest to tone down the conclusions, highlighting that the species is an appropriate candidate for further string-pulling tests.

Also, I am not convinced that the birds understood the task. The fact that most gulls that succeeded on their first attempt did not in the subsequent trials suggests that they did not understand the task and might have solved it by chance (or had some motivational issues which still needs to be discussed).

Finally, I am concerned that the test trials were identical with the last habituation trial.. thus giving the bird opportunity to interact with the proper experimental set-up before the actual test. This is unusual methodology. Usually, prior to the experiment, a short piece of string that cannot be pulled is given to the subjects to prevent a pre-exposure to the actual task but give an opportunity to habituate to the string. I am not convinced by the authors justification for such procedure.

Detailed comments:

Abstract:

l. 16 "Although at least 95 avian species..." in line 44 (introduction) it says 90 avian species. Please correct where appropriate

Introduction:

I enjoyed reading this introduction very much. It is short, well-written, and on the point. I found sentences in ll. 49ff and 58ff slightly contradicting though: one states "waterbirds are not expected to be particularly intelligent because of their small brain size" and the other says "waterbirds are suspected to use cognitively advanced behaviours"...

Methods:

l. 140 The vertical string-pulling test has also been solved by using birds' beak

l.145 How long was the string? Was it standardised across all birds and colonies? From the video it does not look long enough to require more than one pull to access the petridish, making the task even less cognitively demanding compared to the singbird and parrot studies where usually several pulls were necessary, which in turn requires a finer motor coordination.

l. 149, Some of the references appear to be misleading. E.g.:

In Heinrich & Bugnyar 2005 the ravens were given "opportunity to see the string without being allowed to pull it up and/or to use it as a tool to access food".

Also, in Krasheninnikova & Wanker 2010 the habituation involved only empty pieces of strings: "...,"all birds used in the present study had the opportunity to see and interact with the string, as pieces of 3-mm-thick twisted knitting wool strings were hung up in both ends of the aviary at a branch or a perch, where they were freely accessible for all animals."

l. 163 ff I do not understand why did the authors choose to give the birds an opportunity to access the food by pulling the string in the last habitual trial and prior to actual test trials? Also, placing a second piece of food at the edge of the box increases the chance that the task can be solved by local enhancement without any means-end understanding. In my opinion, those are two serious conceptual flaws of the present study. Please give some justification for this methods.

l.223f Again, the first test trial is not the first exposure to the set-up as it is identical to the last habituation trial. This should be taken into account and be reflected in the results.

Discussion:

l.230ff So, what does it mean then if the birds that successfully pulled the string in the last habituation trial failed to do so in the test trials?

l.258ff However, it remains unclear whether or not they show understanding of the task.

l.300 Again, the single string task does not necessarily require advanced cognitive skills.

Decision letter (RSOS-211343.R0)

Dear Miss Lamarre

The Editors assigned to your paper RSOS-211343 "Waterbird solves the cognitive string-pull test" have now received comments from reviewers and would like you to revise the paper in accordance with the reviewer comments and any comments from the Editors. Please note this decision does not guarantee eventual acceptance.

Please submit your revised manuscript and required files (see below) no later than 21 days from today's (ie 29-Sep-2021) date. Note: the ScholarOne system will 'lock' if submission of the revision is attempted 21 or more days after the deadline. If you do not think you will be able to meet this deadline please contact the editorial office immediately.

on behalf of Dr Shinya Yamamoto (Associate Editor) and Essi Viding (Subject Editor)
openscience@royalsociety.org

Associate Editor Comments to Author (Dr Shinya Yamamoto):

Associate Editor: 1

Comments to the Author:

Now we have review comments from two specialists. They have given detailed comments which will be helpful for your revision. Please take them fully into consideration.

Reviewer comments to Author:

Reviewer: 1

Comments to the Author(s)

Review

Manuscript number: RSOS-211343

Title: Waterbird solves the cognitive string-pull test

The authors have written an interesting study about string-pulling performance in wild ring-billed gulls. What I particularly like is the field aspect of this study and the number of individuals tested (124 individuals, four breeding colonies), it sounds like it required significant work and I really appreciate the research effort. Although they are crucial, there are still few experiments assessing problem solving in seabirds, and wild populations more generally. It was also interesting to see that some gulls solved the task while performing incubation duties, which validates the use of such setting for further wild cognitive experiments with this species. There are some points regarding how the results are interpreted which may benefit from greater parsimony though (e.g., as it is, the string-pulling test presented here does not allow assessing

means-end understanding). In my opinion, the authors are testing whether gulls can spontaneously pull on a single rewarded string. Perhaps the authors should introduce the study and discuss the results with this aspect in mind, and with more economy in the explanations. Suggestions are outlined below.

Title

Perhaps replace the term "waterbird" by a term more specific about the species in question (or family). Waterbirds vary significantly in their ecology, morphology, and behaviour (e.g. gulls are extremely different from flamingos, penguins, or pelicans, which are also Aequorlithornithes) I also disagree with "the cognitive" aspect of the string-pull test: pulling on a single rewarded string does not allow such claim

Abstract

Means-end understanding is involved when animals solve different experimental conditions, for instance when they show goal-directedness, rely on proximity, and/ or do not depend on visual feedback (see review of Jacobs and Osvath, 2015).

Introduction

Lines 29-31: I really like this

Lines 35-36: one of the most extensively studied

Lines 37-39: this is an overclaim, it depends on the experimental conditions administered, perhaps be more specific

Lines 41-43: this sentence is not necessary (see Jacobs and Osvath 2015)

Lines 58-59: be careful, tool use is not always cognitively complex, and the interpretations made on puffins have been highly reassessed (e.g. Farrar 2020 and the work that followed e.g. Sándor and Miklósi 2020)

Line 64: it is not yet a cognitive challenge, this is the first interesting step, but more conditions are necessary to be able to safely speak about "cognitive capacities", especially with the use of the horizontal configuration

Line 66: the kagu is not a waterbird

Line 69: perhaps add s at generalist

Methods

Lines 73-78: for the future, it would be interesting to get data on other variables such as sex or personality, in order to interpret whether and how these variables affect performance on the task
Line 80: replace least likely by less likely

Line 113: what do you mean by "privacy"? Other gulls are usually chased by the parents, perhaps you meant privacy from the other partner? If yes, it is the same as the risk for social learning

Lines 113-122: perhaps the fence is too close from the nest? Was there a specific reason for this setting? Perhaps in the future the fence could be one meter away

Also, in video S1 sometimes we have the impression that the gull is a bit disturbed by the box. I understand that it might be difficult to run problem-solving experiments in the field and that the authors wanted to test breeding individuals as it is more convenient, but perhaps in the future you should put the box a bit far away to allow the birds to be more at ease

Lines 140-141: and this horizontal configuration is also considered less cognitively difficult than the vertical one, but of course it is more appropriate for non-perching birds

Line 143: perhaps "configuration" instead of rendition?

Lines 150-154: it's just my opinion, but I would test each nest once at a time. If a problem occurs with the apparatus, for instance if a string detaches from the box accidentally, it is usually safer to focus observations on one nest

Lines 158-167: I appreciate the fact that authors first ensured to teach the gulls to associate the food with the string. Perhaps another learning phase might have been interesting to add after the habituation phase. Often animals are taught to pull on the string too, which represents a

necessary step to proceed to further, more complex experimental string-pulling conditions (see the variety of conditions shown in the review made by Jacobs and Osvath 2015).

Lines 193-195: the gull did not attempt “to solve the task”, perhaps reformulate by “the gull was attempting to reach the food reward if it...”

Lines 205-206: there could be low-level social learning mechanisms in play (for instance social motivation, social enhancement) because the birds interacted with the box, perhaps be more specific

Discussion

Lines 264-265: stay parsimonious, you did not relate your finding to other cognitive tasks or behavioural traits, and did not use various two-strings experimental conditions

Line 270: beware, this study tested a very limited number of captive individuals

Line 276: add the reference in parentheses

Lines 283-284: perhaps cognitive abilities may be selected also within the sexual pair

Line 300: perhaps a more cautious sentence is required, see my comment above

Lines 312-313: explain why it is important to study other bird species outside passerine and psittacine species (from the fundamental cognitive point of view, not for the conservation one)

Line 330: be more cautious

References

Use italics for latin names and recheck the writing of the references (e.g. 4)

Video S1: correct the legend sentence shown in the movie

Reviewer: 2

Comments to the Author(s)

The authors present a nice study on string-pulling in a previously untested species of waterbirds. The authors employed a widely used string-pulling paradigm to assess the problem-solving abilities of wild ring-billed gulls. Overall, this is an interesting work and I think the paper would make a valuable contribution to the field as it adds important data to body of literature on string-pulling that has been heavily dominated by work on Passeriformes and Psittaciformes. However, there are a few aspects of the paper that I would like to see clarified / improved.

My main concern is that in its current form, the manuscript creates an impression that successful performance in a single string task is still commonly considered to show means-end understanding, which is not the case. It is widely accepted that a successful pulling up a string only proves that a species possesses the required sensorimotor capacities and are therefore suitable for further testing as it is unclear what cognitive abilities underpin a successful response. The basic string-pulling skills do not really say much about cognitive skills without other string-pulling tasks testing for causal understanding, or connectivity skills. Thus I would suggest to tone down the conclusions, highlighting that the species is an appropriate candidate for further string-pulling tests.

Also, I am not convinced that the birds understood the task. The fact that most gulls that succeeded on their first attempt did not in the subsequent trials suggests that they did not understand the task and might have solved it by chance (or had some motivational issues which still needs to be discussed).

Finally, I am concerned that the test trials were identical with the last habituation trial.. thus giving the bird opportunity to interact with the proper experimental set-up before the actual test. This is unusual methodology. Usually, prior to the experiment, a short piece of string that cannot be pulled is given to the subjects to prevent a pre-exposure to the actual task but give an

opportunity to habituate to the string. I am not convinced by the authors justification for such procedure.

Detailed comments:

Abstract:

l. 16 “Although at least 95 avian species...” in line 44 (introduction) it says 90 avian species. Please correct where appropriate

Introduction:

I enjoyed reading this introduction very much. It is short, well-written, and on the point. I found sentences in ll. 49ff and 58ff slightly contradicting though: one states “waterbirds are not expected to be particularly intelligent because of their small brain size” and the other says “waterbirds are suspected to use cognitively advanced behaviours”...

Methods:

l. 140 The vertical string-pulling test has also been solved by using birds’ beak

l.145 How long was the string? Was it standardised across all birds and colonies? From the video it does not look long enough to require more than one pull to access the petridish, making the task even less cognitively demanding compared to the singbird and parrot studies where usually several pulls were necessary, which in turn requires a finer motor coordination.

l. 149, Some of the refences appear to be misleading. E.g.:

In Heinrich & Bugnyar 2005 the ravens were given “opportunity to see the string without being allowed to pull it up and/or to use it as a tool to access food”.

Also, in Krasheninnikova & Wanker 2010 the habituation involved only empty pieces of strings: ...“all birds used in the present study had the opportunity to see and interact with the string, as pieces of 3-mm-thick twisted knitting wool strings were hung up in both ends of the aviary at a branch or a perch, where they were freely accessible for all animals.”

l. 163 ff I do not understand why did the authors choose to give the birds an opportunity to access the food by pulling the string in the last habitual trial and prior to actual test trials? Also, placing a second piece of food at the edge of the box increases the chance that the task can be solved by local enhancement without any means-end understanding. In my opinion, those are two serious conceptual flaws of the present study. Please give some justification for this methods.

l.223f Again, the first test trial is not the first exposure to the set-up as it is identical to the last habituation trial. This should be taken into account and be reflected in the results.

Discussion:

ll.230ff So, what does it mean then if the birds that successfully pulled the string in the last habituation trial failed to do so in the test trials?

l.258ff However, it remains unclear whether or not they show understanding of the task.

l.300 Again, the single string task does not necessarily require advanced cognitive skills.

===PREPARING YOUR MANUSCRIPT===

===PREPARING YOUR REVISION IN SCHOLARONE===

<https://royalsociety.org/journals/authors/author-guidelines/#supplementary-material> to include a suitable title and informative caption. An example of appropriate titling and captioning may be found at https://figshare.com/articles/Table_S2_from_Is_there_a_trade-off_between_peak_performance_and_performance_breadth_across_temperatures_for_aerobic_sc_ope_in_teleost_fishes_/3843624.

Author's Response to Decision Letter for (RSOS-211343.R0)

See Appendix A.

RSOS-211343.R1 (Revision)

Review form: Reviewer 1

Is the manuscript scientifically sound in its present form?

Yes

Are the interpretations and conclusions justified by the results?

Yes

Is the language acceptable?

Yes

Do you have any ethical concerns with this paper?

No

Have you any concerns about statistical analyses in this paper?

No

Recommendation?

Accept with minor revision (please list in comments)

Comments to the Author(s)

Review

Manuscript number: RSOS-211343

Title: Waterbird solves the cognitive string-pull test

The authors have properly nuanced their interpretations and conclusions. I just have some remarks about certain aspects of the manuscript.

Abstract

Line 23: regarding your sentence "Ring-billed gulls are thus the first waterbird known to succeed at string-pulling", could you be more specific? Said this way this may lead to confusion (subjects have passed some string-pulling conditions), perhaps it is better to say that gulls succeed at pulling on horizontal, single rewarded strings.

Introduction

Lines 37-41: perhaps always employ "configuration" for horizontal/vertical string, and "condition" for basic (single rewarded string), crossed, lack of contact etc. conditions, and try to stay consistent throughout the manuscript. So here, replace "for the most complex configurations" by "for the most complex conditions".

Also perhaps reformulate the sentence to make it clearer for the reader (it's a bit long).

Line 42: provide more than one example in brackets

Line 49: correct: Charadriiformes not Charadriformes

Line 131: content of Figure 2 does not appear correctly in the manuscript

Line 251-252: try to stay consistent in throughout the manuscript, either use number or text e.g. you wrote "4 of them" but above you use text for numbers

Line 263: please be more specific, see my comment above, you should specify horizontal configuration and only one "basic" condition (often not used during testing, but during training).

Line 288-289: perhaps do not speak about means-end understanding now, just say the importance of administering further conditions to properly assess the mechanisms at play

Line 292-293: reformulate, see my comment above

Line 341: "We suggest that various configurations of the string-pull test outlined here should be administered"?

Line 344: remove "avian and"

Review form: Reviewer 2**Is the manuscript scientifically sound in its present form?**

Yes

Are the interpretations and conclusions justified by the results?

Yes

Is the language acceptable?

Yes

Do you have any ethical concerns with this paper?

No

Have you any concerns about statistical analyses in this paper?

No

Recommendation?

Accept as is

Comments to the Author(s)

The authors have dealt with all comments raised in the first review. I thank the authors for providing a very careful revision and a very detailed point-by-point reply. I am fully satisfied with the revised version and have no more points to raise.

Decision letter (RSOS-211343.R1)

Dear Miss Lamarre

On behalf of the Editors, we are pleased to inform you that your Manuscript RSOS-211343.R1 "Waterbird solves the string-pull test" has been accepted for publication in Royal Society Open Science subject to minor revision in accordance with the referees' reports. Please find the referees' comments along with any feedback from the Editors below my signature.

Please submit your revised manuscript and required files (see below) no later than 7 days from today's (ie 01-Nov-2021) date. Note: the ScholarOne system will 'lock' if submission of the revision is attempted 7 or more days after the deadline. If you do not think you will be able to meet this deadline please contact the editorial office immediately.

on behalf of Dr Shinya Yamamoto (Associate Editor) and Essi Viding (Subject Editor)

Associate Editor Comments to Author (Dr Shinya Yamamoto):

Associate Editor: 1

Comments to the Author:

Both reviewers are satisfied with the revision and I agree with them. Please amend the points the reviewer 1 raised. Thank you for your submission of the well-written revision.

Reviewer comments to Author:

Reviewer: 2

Comments to the Author(s)

The authors have dealt with all comments raised in the first review. I thank the authors for providing a very careful revision and a very detailed point-by-point reply. I am fully satisfied with the revised version and have no more points to raise.

Reviewer: 1

Comments to the Author(s)

Review

Manuscript number: RSOS-211343

Title: Waterbird solves the cognitive string-pull test

The authors have properly nuanced their interpretations and conclusions. I just have some remarks about certain aspects of the manuscript.

Abstract

Line 23: regarding your sentence "Ring-billed gulls are thus the first waterbird known to succeed at string-pulling", could you be more specific? Said this way this may lead to confusion (subjects have passed some string-pulling conditions), perhaps it is better to say that gulls succeed at pulling on horizontal, single rewarded strings.

Introduction

Lines 37-41: perhaps always employ "configuration" for horizontal/vertical string, and "condition" for basic (single rewarded string), crossed, lack of contact etc. conditions, and try to stay consistent throughout the manuscript. So here, replace "for the most complex configurations" by "for the most complex conditions".

Also perhaps reformulate the sentence to make it clearer for the reader (it's a bit long).

Line 42: provide more than one example in brackets

Line 49: correct: Charadriiformes not Charadriformes

Line 131: content of Figure 2 does not appear correctly in the manuscript

Line 251-252: try to stay consistent in throughout the manuscript, either use number or text e.g. you wrote "4 of them" but above you use text for numbers

Line 263: please be more specific, see my comment above, you should specify horizontal configuration and only one "basic" condition (often not used during testing, but during training).

Line 288-289: perhaps do not speak about means-end understanding now, just say the importance of administering further conditions to properly assess the mechanisms at play

Line 292-293: reformulate, see my comment above

Line 341: "We suggest that various configurations of the string-pull test outlined here should be administered"?

Line 344: remove "avian and"

===PREPARING YOUR MANUSCRIPT===

one version should clearly identify all the changes that have been made (for instance, in coloured highlight, in bold text, or tracked changes);

===PREPARING YOUR REVISION IN SCHOLARONE===

-- Ensure that your data access statement meets the requirements at <https://royalsociety.org/journals/authors/author-guidelines/#data>. You should ensure that you cite the dataset in your reference list. If you have deposited data etc in the Dryad repository, please only include the 'For publication' link at this stage. You should remove the 'For review' link.

-- If you are requesting an article processing charge waiver, you must select the relevant waiver option (if requesting a discretionary waiver, the form should have been uploaded, see 'File upload' above).

-- If you have uploaded any electronic supplementary (ESM) files, please ensure you follow the guidance at <https://royalsociety.org/journals/authors/author-guidelines/#supplementary-material> to include a suitable title and informative caption. An example of appropriate titling and captioning may be found at https://figshare.com/articles/Table_S2_from_Is_there_a_trade-off_between_peak_performance_and_performance_breadth_across_temperatures_for_aerobic_scope_in_teleost_fishes_/3843624.

Author's Response to Decision Letter for (RSOS-211343.R1)

See Appendix B.

Decision letter (RSOS-211343.R2)

Dear Miss Lamarre,

I am pleased to inform you that your manuscript entitled "Waterbird solves the string-pull test" is now accepted for publication in Royal Society Open Science.

The proof of your paper will be available for review using the Royal Society online proofing system and you will receive details of how to access this in the near future from our production

office (openscience_proofs@royalsociety.org). We aim to maintain rapid times to publication after acceptance of your manuscript and we would ask you to please contact both the production office and editorial office if you are likely to be away from e-mail contact to minimise delays to publication. If you are going to be away, please nominate a co-author (if available) to manage the proofing process, and ensure they are copied into your email to the journal.

on behalf of Dr Shinya Yamamoto (Associate Editor) and Essi Viding (Subject Editor)
openscience@royalsociety.org

Appendix A

re: ms RSOS-211343, Waterbird solves the cognitive string-pull test

Dear Drs. Yamamoto and Viding,

Thank you for providing us the opportunity to revise our manuscript for possible publication in *Royal Society Open Science*. We found the comments by the two reviewers to be very thoughtful and constructive. We have revised our manuscript in light of these comments, and explained our changes point by point in detail below. We have uploaded one copy of the manuscript with the changes highlighted, plus a clean version without the highlights. Line numbers in our responses refer to those in the highlighted version (tracked changes inline).

XX

Editor: Now we have review comments from two specialists. They have given detailed comments which will be helpful for your revision. Please take them fully into consideration.

Response: Thank you for providing us the opportunity to revise our manuscript. We address all of the reviewers' concerns below.

XX

Reviewer 1: The authors have written an interesting study about string-pulling performance in wild ring-billed gulls. What I particularly like is the field aspect of this study and the number of individuals tested (124 individuals, four breeding colonies), it sounds like it required significant work and I really appreciate the research effort. Although they are crucial, there are still few experiments assessing problem solving in seabirds, and wild populations more generally. It was also interesting to see that some gulls solved the task while performing incubation duties, which validates the use of such setting for further wild cognitive experiments with this species.

Response: We thank the reviewer for their kind comments about our research.

Reviewer 1: There are some points regarding how the results are interpreted which may benefit from greater parsimony though (e.g., as it is, the string-pulling test presented here does not allow assessing means-end understanding). In my opinion, the authors are testing whether gulls can spontaneously pull on a single rewarded string. Perhaps the authors should introduce the study and discuss the results with this aspect in mind, and with more economy in the explanations. Suggestions are outlined below.

Response: We agree that our interpretation of our results would benefit from being more nuanced and have therefore included caveats throughout our text. We have removed any mention that solving our configuration of the string-pull test demonstrated advanced cognitive skills (Title, L14, L37-39, L69, L331) and we clarify that our subjects were only presented with one configuration of the string-pull test (L20, L68, L280, L293, L340-341). We also acknowledge that few gulls that solved the test were then

presented with a subsequent opportunity to solve the test, thus limiting our ability to analyze the repeatability of test success. We now discuss this limitation and acknowledge that it prevents us from making inferences about the solvers' comprehension of the test (L283-291). We also caution that more testing than a single iteration of the string-pull test is required to properly assess the cognitive abilities of an animal (L292-296).

Reviewer 1: Specific Comments: Title

Perhaps replace the term "waterbird" by a term more specific about the species in question (or family). Waterbirds vary significantly in their ecology, morphology, and behaviour (e.g. gulls are extremely different from flamingos, penguins, or pelicans, which are also Aequorlithornithes)
I also disagree with "the cognitive" aspect of the string-pull test: pulling on a single rewarded string does not allow such claim

Response: We have removed the term "cognitive" from the title. We chose the term waterbird as this is the broadest taxonomic group to which ring-billed gulls belong and for which no species had previously been tested with the string-pull test. Using the word waterbird also brings forward the lack of attention that birds pertaining to this clade have received regarding their cognitive abilities. We specify the species tested in the abstract.

Reviewer 1: Specific Comments: Abstract

Means-end understanding is involved when animals solve different experimental conditions, for instance when they show goal-directedness, rely on proximity, and/ or do not depend on visual feedback (see review of Jacobs and Osvath, 2015).

Response: We agree that testing for means-end understanding requires further experimental conditions than what we have assessed in our experiment and we have removed this wording for our abstract (L14).

Reviewer 1: Specific Comments: Introduction

Lines 29-31: I really like this

Response: We thank the reviewer for their kind comment.

Reviewer 1: Lines 35-36: one of the most extensively studied

Response: We agree and have changed the wording accordingly (L35).

Reviewer 1: Lines 37-39: this is an overclaim, it depends on the experimental conditions administered, perhaps be more specific

Response: We have reworded this sentence to specify that insight and means-end understanding are tested with the most complicated configurations of the string-pull tests, which usually require the animal to choose the correct string among an array of choices (L37-39).

Reviewer 1: Lines 41-43: this sentence is not necessary (see Jacobs and Osvath 2015)

Response: We agree and we have reworded the sentence to remove the duplication in information from the Jacobs and Osvath 2015 paper (L43-46).

Reviewer 1: Lines 58-59: be careful, tool use is not always cognitively complex, and the interpretations made on puffins have been highly reassessed (e.g. Farrar 2020 and the work that followed e.g. Sándor and Miklósi 2020)

Response: We agree and have reworded the sentence to imply that tool use is sometimes perceived as a cognitively advanced behaviour. We also agree that the evidence of tool-use in puffins is not yet convincing and have thus removed this mention from our text (L61-63).

Reviewer 1: Line 64: it is not yet a cognitive challenge, this is the first interesting step, but more conditions are necessary to be able to safely speak about “cognitive capacities”, especially with the use of the horizontal configuration

Response: We have clarified here that ring-billed gulls were only presented with one iteration of the string-pull test (L68) and we have replaced the wording “cognitive challenge” with “puzzle” (L69) as a more accurate description of the task they undertook.

Reviewer 1: Line 66: the kagu is not a waterbird

Response: We agree that the kagu is a terrestrial bird, but include it here because the most recent phylogenetic analyses include it in the family Rhynochetidae, which is part of the order Eurypygiformes and the broader Aequorlitorornithes (waterbird) clade. We feel it would be inaccurate to exclude this species from our taxonomic description of the waterbird clade and have instead added a clarification that this is a terrestrial species in an otherwise waterbird clade (L71).

Reviewer 1: Line 69: perhaps add s at generalist

Response: We have adjusted the sentence accordingly (L75).

Reviewer 1: Specific Comments: Methods

Lines 73-78: for the future, it would be interesting to get data on other variables such as sex or personality, in order to interpret whether and how these variables affect performance on the task

Response: We agree and note this research was only meant to determine whether ring-billed gulls can solve the string-pull test. Our future research will dive deeper into the factors that affect their performance at the test now that we know that this is a suitable task for this species to solve.

Reviewer 1: Line 80: replace least likely by less likely

Response: We have changed the wording accordingly (L86).

Reviewer 1: Line 113: what do you mean by "privacy"? Other gulls are usually chased by the parents, perhaps you meant privacy from the other partner? If yes, it is the same as the risk for social learning

Response: We used the fences as a way to prevent harassment from neighbouring gulls that would otherwise have seen food available at the subject's nest and possibly attempted to steal it. We had observed pilfering prior to this study, even when the parents were present at the nest, and we thus used the fences as a way to minimize this issue. We have adjusted our wording on L119-120 to better clarify that fences were used to prevent pilfering by neighbouring gulls.

Reviewer 1: Lines 113-122: perhaps the fence is too close from the nest? Was there a specific reason for this setting? Perhaps in the future the fence could be one meter away

Response: While it would have been ideal to provide each pair with a wide enclosure around their nest, we were limited in space given the proximity of neighbouring nests (typically within a 0.6-0.7m radius). We chose a square arrangement of 1.3x1.3m as a compromise between ensuring that the parents' wingspan (usually ~120-125cm) would fit within the fenced enclosure while also not encroaching on the territory of neighboring gulls. We initially kept the burlap fences low to facilitate the return of parents to their nest after the disturbance of setting up the fences (hammering posts in the ground and stapling the burlap around them). We confirmed that parents returned to their nest within a few minutes' time if the burlap was initially kept low and that the 1.3x1.3m raised fences did not impair their ability to fly in and out of the enclosure. We have added those clarifications on L126-129.

Reviewer 1: Also, in video S1 sometimes we have the impression that the gull is a bit disturbed by the box. I understand that it might difficult to run problem-solving experiments in the field and that the authors wanted to test breeding individuals as it is more convenient, but perhaps in the future you should put the box a bit far away to allow the birds to be more at ease

Response: This is a good point and we take note of this for our future methods. We had chosen to place the testing box at the rim of the gulls' nest to avoid a low rate of participation in interacting with the test. Due to the gulls' high drive to incubate, we were concerned that placing the box away from the nest would discourage the birds from attempting to solve the test, especially if that meant that they could not sit on their eggs at the same time. We note that the vast majority of our subjects were not in contact with the box while incubating, but nevertheless plan to include a greater distance between the apparatus and the nests to avoid this issue in the future.

Reviewer 1: Lines 140-141: and this horizontal configuration is also considered less cognitively difficult than the vertical one, but of course it is more appropriate for non-perching birds.

Response: We agree and we clarify that this configuration of the string-pull test is less difficult to solve since the string does not need to be held in place between pulls (L149-151).

Reviewer 1: Line 143: perhaps “configuration” instead of rendition?

Response: We have adjusted the text accordingly (L151).

Reviewer 1: Lines 150-154: it’s just my opinion, but I would test each nest once at a time. If a problem occurs with the apparatus, for instance if a string detaches from the box accidentally, it is usually safer to focus observations on one nest

Response: We agree that testing one nest at a time would be ideal. We were however restricted in the time we could spend at each colony since we targeted gulls during their last week of incubation when they are most tolerant of disturbance. This gave us only 5-7 days to spend at each site, and 2 of those days were dedicated to catching birds. We were fortunate to not have encountered any issues with our apparatus.

Reviewer 1: Lines 158-167: I appreciate the fact that authors first ensured to teach the gulls to associate the food with the string. Perhaps another learning phase might have been interesting to add after the habituation phase. Often animals are taught to pull on the string too, which represents a necessary step to proceed to further, more complex experimental string-pulling conditions (see the variety of conditions shown in the review made by Jacobs and Osvath 2015).

Response: We thank the reviewer for their kind comment. If we had had more time to remain at each colony before needing to move on to the next one, we would have considered implementing a broader cognitive challenge where the birds were also presented with a baited elastic string as a way to mimic the vertical test where birds need to stop the string from being dragged backwards between pulls.

Reviewer 1: Lines 193-195: the gull did not attempt “to solve the task”, perhaps reformulate by “the gull was attempting to reach the food reward if it...”

Response: We agree and we have changed the wording accordingly to improve accuracy (L207).

Reviewer 1: Lines 205-206: there could be low-level social learning mechanisms in play (for instance social motivation, social enhancement) because the birds interacted with the box, perhaps be more specific

Response: We have changed the wording to clarify that the presence of both mates at the nest did not appear to enhance the success rate of these individuals at the string-pull test (220-221). We acknowledge however that we cannot exclude the possibility of a leveling down effect where the actions of the first unsuccessful gulls were repeated by their mates, which were equally unsuccessful. Given that there were only three instances where both mates were present during a test trial (all during the last

trial) and where both interacted with the box, we consider that our results are accurately representative of the success rate of ring-billed gulls, and, if anything, are likely somewhat conservative.

Reviewer 1: Specific Comments: Discussion

Lines 264-265: stay parsimonious, you did not relate your finding to other cognitive tasks or behavioural traits, and did not use various two-strings experimental conditions

Response: We agree and we have adjusted the wording to describe our results as the first to demonstrate a waterbird being capable of solving a configuration of the string-pull test (L292-293).

Reviewer 1: Line 270: beware, this study tested a very limited number of captive individuals

Response: This is a good point and we have included in our manuscript that this study refers to a very small sample size of 5 individuals (L300-301).

Reviewer 1: Line 276: add the reference in parentheses

Response: We have added two references at the end of L306 (Emery 2006 and Sol 2009) that support our claim that avian cognition is associated with a bird's life history and with its foraging flexibility.

Reviewer 1: Lines 283-284: perhaps cognitive abilities may be selected also within the sexual pair

Response: We agree that there is some evidence for assortative mating for cognitive abilities and for mate preference for more intelligent individuals, and have thus included this fascinating possibility in the text (L314-315).

Reviewer 1: Line 300: perhaps a more cautious sentence is required, see my comment above

Response: We have revised this sentence to remove the implications that solving the string-pull test equates to advanced cognitive skills (L331).

Reviewer 1: Lines 312-313: explain why it is important to study other bird species outside passerine and psittacine species (from the fundamental cognitive point of view, not for the conservation one)

Response: This is a good point and we have added a sentence that address why continuing to explore the cognitive abilities of non-passerine, non-psittacine species are important for comparative studies of cognition in light of habitat requirements (L344-346).

Reviewer 1: Line 330: be more cautious

Response: We agree and we have replaced the wording “attempted to solve the test” with “attempted to reach the food reward” (L364).

Reviewer 1: References: Use italics for latin names and recheck the writing of the references (e.g. 4)

Response: We have revised the literature cited section to italicize all latin names and to ensure that the correct citation format has been respected throughout.

Reviewer 1: Video S1: correct the legend sentence shown in the movie

Response: We have corrected the legend shown in the first clip of Video S1 where “unsuccessful” was misspelled.

XX

Reviewer 2: The authors present a nice study on string-pulling in a previously untested species of waterbirds. The authors employed a widely used string-pulling paradigm to assess the problem-solving abilities of wild ring-billed gulls. Overall, this is an interesting work and I think the paper would make a valuable contribution to the field as it adds important data to body of literature on string-pulling that has been heavily dominated by work on Passeriformes and Psittaciformes. However, there are a few aspects of the paper that I would like to see clarified / improved.

Response: We thank the reviewer for their kind appraisal of our manuscript. We respond to the specific requests for clarification below.

Reviewer 2: My main concern is that in its current form, the manuscript creates an impression that successful performance in a single string task is still commonly considered to show means-end understanding, which is not the case. It is widely accepted that a successful pulling up a string only proves that a species possesses the required sensorimotor capacities and are therefore suitable for further testing as it is unclear what cognitive abilities underpin a successful response. The basic string-pulling skills do not really say much about cognitive skills without other string-pulling tasks testing for causal understanding, or connectivity skills. Thus I would suggest to tone down the conclusions, highlighting that the species is an appropriate candidate for further string-pulling tests.

Response: We agree that a single configuration of the string-pull test provides limited understanding of an animal’s cognitive abilities, which are better tested using a range of string-pull test iterations. We also note that reviewer 1 shared this concern. In response to both reviewers, we have included caveats to better portray our research as a step towards showing that ring-billed gulls (and waterbirds in general) are promising candidates for the study of avian cognition while also cautioning that additional testing other than a single iteration of the string-pull test is required to properly assess the cognitive abilities of animals (L293-296). As mentioned above, we have removed any mention that solving our configuration of the string-pull test demonstrated advanced cognitive skills (Title, L14, L37-39, L69, L331) and we clarify that our subjects were only presented with one configuration of the string-pull test (L20, L68, L280, L293, L340-341).

Reviewer 2: Also, I am not convinced that the birds understood the task. The fact that most gulls that succeeded on their first attempt did not in the subsequent trials suggests that they did not understand the task and might have solved it by chance (or had some motivational issues which still needs to be discussed).

Response: We agree that we had overlooked discussing the low repeatability of test success among the few birds that underwent a subsequent trial after an initial success at the string-pull test. We now include a section in the discussion where we discuss the possibility that some solvers did not understand the task, as was evidenced by their subsequent failure to solve the test L283-291.

Reviewer 2: Finally, I am concerned that the test trials were identical with the last habituation trial.. thus giving the bird opportunity to interact with the proper experimental set-up before the actual test. This is unusual methodology. Usually, prior to the experiment, a short piece of string that cannot be pulled is given to the subjects to prevent a pre-exposure to the actual task but give an opportunity to habituate to the string. I am not convinced by the authors justification for such procedure.

Response: This is a good point. We chose to present the birds with the complete experimental set-up, albeit with a lidless top, due to the limited number of opportunities we had to deploy the string-pull test. Although this species is usually very tolerant of disturbance at their colonies, we were concerned that the addition of novelty during testing (after having already captured one parent per pair, set up a fence around their nest, and added elements of the testing apparatus, all within a span of 3-4 days) would deter them from engaging with the string-pull test within the 10min trials. We have added this justification to our methods (L174-176) and results (L247-249). That said, given that only eight gulls pulled on the string during the fifth habituation trial (including 3 gulls that were never present during subsequent string-pull test trials), and that the rest of them ignored the string and took the sausage directly through the lidless top, we are confident that the overall string-pull test results were not skewed by this prior experience with the string and the petri dish.

Reviewer 2: Abstract:

I. 16 “Although at least 95 avian species...” in line 44 (introduction) it says 90 avian species. Please correct where appropriate

Response: We thank the reviewer for picking up on this error and we have adjusted L16 of the abstract to 90 species.

Reviewer 2: Introduction:

I enjoyed reading this introduction very much. It is short, well-written, and on the point. I found sentences in ll. 49ff and 58ff slightly contradicting though: one states “waterbirds are not expected to be particularly intelligent because of their small brain size” and the other says “waterbirds are suspected to use cognitively advanced behaviours”...

Response: We thank the reviewer for their kind comment about our introduction. We changed our wording at L53-54 to clarify that cognitive research has perhaps overlooked waterbirds because their

relative brain size is smaller than corvids and parrots', thus they were not considered as cognitively interesting to study as bigger brained birds.

Reviewer 2: Methods:

l. 140 The vertical string-pulling test has also been solved by using birds' beak

Response: We agree and we have adjusted the text accordingly (L147).

Reviewer 2: l.145 How long was the string? Was it standardised across all birds and colonies? From the video it does not look long enough to require more than one pull to access the petridish, making the task even less cognitively demanding compared to the singbird and parrot studies where usually several pulls were necessary, which in turn requires a finer motor coordination.

Response: The string was 20cm long and, for all trials/nests/colonies, the petri dish was placed in the box 10cm away from the slit so that 10 cm of string extended outside of the box (L170-171). We acknowledge that the video camera's perspective sometimes obscured these distances. Prior to deploying this experiment, we tested the string-pull test apparatus on foraging gulls outside of a colony setting and on incubating gulls that were not part of our sample. We found that incubating gulls were not interested in attempting to get the food reward (did not interact with the box in any way) when the petri dish containing the sausage was placed at the back of the box, a placement that would have required multiple pulls on the string to extract the food from the box. Instead, we found that placing the petri dish 10cm away from the open slit prevented the gulls from accessing it directly with their beak while keeping them motivated to continue interacting with the box in an attempt to obtain the food reward. We did not have this problem with free-foraging gulls which always spent more time interacting with the box and which were also more aggressive with it in general, to the point of thrashing the box around – a behaviour never once seen with incubating birds. Because of this lack of participation from incubating gulls when the petri dish was too far back, we opted to place it 10cm away from the opening slit. Although one pull should have been enough to extract the petri dish, in most cases the birds had to pull on the string several times as the petri dish tended to be pushed back in the box before being extracted (as seen in Movie S2). We added the specifications regarding the placement of the petri dish at L154-155. The length of the string protruding out of the box is indicated on L171.

Reviewer 2: l. 149, Some of the references appear to be misleading. E.g.:

In Heinrich & Bugnyar 2005 the ravens were given "opportunity to see the string without being allowed to pull it up and/or to use it as a tool to access food".

Also, in Krasheninnikova & Wanker 2010 the habituation involved only empty pieces of strings: ..., "all birds used in the present study had the opportunity to see and interact with the string, as pieces of 3-mm-thick twisted knitting wool strings were hung up in both ends of the aviary at a branch or a perch, where they were freely accessible for all animals."

Response: We agree that the Heinrich & Bugnyar 2005 and Krasheninnikova & Wanker 2010 references used as evidence that habituation is common practice in string-pull test studies inaccurately implied that we implemented a habituation protocol similar to theirs. In comparison, our third reference from this sentence (Auersperg et al. 2011) used a familiarization trial akin to ours where the birds could reach into the test box directly to access the food reward. To avoid misrepresenting the habituation trials used by

Heinrich & Bugnyar 2005 and Krasheninnikova & Wanker 2010, we removed these references from our citation. We kept the Auersperg et al. 2011 citation and added a reference by Wang et al. 2019 since the latter also used a training phase similar to ours where they provided easily accessible food at the slit of their box to habituate the birds to associate their string-pull test apparatus with food (L159-160).

Reviewer 2: l. 163 ff I do not understand why did the authors choose to give the birds an opportunity to access the food by pulling the string in the last habitual trial and prior to actual test trials? Also, placing a second piece of food at the edge of the box increases the chance that the task can be solved by local enhancement without any means-end understanding. In my opinion, those are two serious conceptual flaws of the present study. Please give some justification for this methods.

Response: This is a good point. We chose to give the birds access to every part of the string-pull test apparatus ahead of testing them in an effort to remove all potential neophobic aspects. We had a limited number of opportunities to deploy the string-pull test at each nest and given that either one of the mates could be at the nest at the time of habituation and during the test trials, we sought to minimize any novelties that could deter the birds from engaging with the apparatus. As such, we provided them access to the string during the fifth habituation trial so that it would not be novel to them. We have added this justification to our methods (L174-176). Most birds ignored the string during this last habituation trial and took the sausage from the top of the lidless box.

We continued to place a piece of sausage at the opening slit of the box to ensure that the birds being tested were interested in the food reward in the first place and to induce them to continue exploring the box to obtain the other identical piece of food inside of it. Some birds completely ignored the easily accessible sausage, which made it clear that they either did not recognize it as food or were not interested in eating it. Furthermore, since we tested wild birds in their natural setting, it was not possible to control for food drive by withdrawing feeds from them. We thus used the easily accessible sausage as a way to easily determine whether or not they were interested in obtaining the food in the first place. We have added this clarification to our methods on L186-190.

As shown in the videos, the birds were very precise when grabbing the string or the sausage at the slit, such that we only came across one bird that accidentally pulled on the string while grabbing onto the sausage at the slit during testing. This occurred during the fifth habituation trial and this bird continued to pull on the string afterwards before being able to access the food reward (Movie S3, LP nest 13 unbanded mate at 0:51min). Despite this mitigated success, it was unsuccessful at solving the test during a string-pull test trial and was thus not included in our count of successful birds. Nonetheless, we have added a caveat to the solving success of this particular habituation trial at L253-255. We also note that the opposite situation occurred, where a gull pulled on the string and knocked the accessible sausage out of the slit with the petri dish (Movie S2: LP nest 5 unbanded mate at 2:45min, SB nest 12 unbanded mate at 6:30min, SB nest 32 1015-09604 at 7:10min; Movie S3: OP nest 32 1015-09633 at 3:25min). We considered those to be successes.

Reviewer 2: l.223f Again, the first test trial is not the first exposure to the set-up as it is identical to the last habituation trial. This should be taken into account and be reflected in the results.

Response: We agree that the last habituation trial could be considered an exposure to the string-pull test since it was possible for the gulls to pull the food reward out of the box using the string. However, in

most cases, the gulls still chose to grab the food from the lidless top rather than investigate the string. Except for the eight gulls that pulled the string during the final habituation trial (described on L245-255), the gulls ignored the string and took the sausage directly through the lidless top. We therefore consider that the string-pull test is different enough from the last habituation trial to be evaluated separately. We have added this justification on L247-249 before presenting the results from the last habituation trial.

Reviewer 2: Discussion:

l1.230ff So, what does it mean then if the birds that successfully pulled the string in the last habituation trial failed to do so in the test trials?

Response: As mentioned above, we now include a section in the discussion where we discuss the possibility that some solvers did not understand the task, as evidenced by their subsequent failure to solve the test L283-291.

Reviewer 2: l.258ff However, it remains unclear whether or not they show understanding of the task.

Response: We agree and we have added a section pertaining to this uncertainty at L283-291.

Reviewer 2: l.300 Again, the single string task does not necessarily require advanced cognitive skills

Response: We agree and we have reworded this sentence to remove the implication pertaining to advanced cognitive skills (331).

XX

In addition to those changes outlined above, we have thanked the reviewers in the acknowledgments for their valuable feedback. We also confirm that the manuscript complies with all formatting instructions and we have uploaded the table and figures as standalone files, as requested. We thank you for considering our revised manuscript for publication in *Royal Society Open Science*, and look forward to hearing back from you.

Sincerely,

On behalf of Jessika Lamarre and David Wilson

Appendix B

re: ms RSOS-211343, Waterbird solves the string-pull test

Dear Drs. Yamamoto and Viding,

Thank you for accepting our manuscript for publication in *Royal Society Open Science*, subject to minor revision. We found the comments by reviewer one to be very thoughtful and constructive. We have revised our manuscript in light of these comments, and explained our changes point by point in detail below. We have uploaded one copy of the manuscript with the changes from our first revision highlighted, plus a clean version without the highlights. Line numbers in our responses refer to those in the highlighted version (tracked changes inline).

XX

Editor: Both reviewers are satisfied with the revision and I agree with them. Please amend the points the reviewer 1 raised. Thank you for your submission of the well-written revision.

Response: Thank you for your kind comments and for providing us the opportunity to revise our manuscript a second time. We address all of the reviewers' concerns below.

XX

Reviewer 2: The authors have dealt with all comments raised in the first review. I thank the authors for providing a very careful revision and a very detailed point-by-point reply. I am fully satisfied with the revised version and have no more points to raise.

Response: We thank the reviewer for their kind comments about our revisions, and for the time and consideration they devoted to our manuscript throughout review.

XX

Reviewer 1: The authors have properly nuanced their interpretations and conclusions. I just have some remarks about certain aspects of the manuscript.

Response: We thank the reviewer for their kind appraisal of our revisions. We respond to the specific remarks below.

Reviewer 1: Abstract:

Line 23: regarding your sentence "Ring-billed gulls are thus the first waterbird known to succeed at string-pulling", could you be more specific? Said this way this may lead to confusion (subjects have passed some string-pulling conditions), perhaps it is better to say that gulls succeed at pulling on horizontal, single rewarded strings.

Response: We agree and we have changed the wording according to the reviewer's suggestion to "Ring-billed gulls are thus the first waterbird known to solve a horizontal single-string-rewarded string-pull test" on L23.

Reviewer 1: Introduction:

Lines 37-41: perhaps always employ "configuration" for horizontal/vertical string, and "condition" for basic (single rewarded string), crossed, lack of contact etc. conditions, and try to stay consistent throughout the manuscript. So here, replace "for the most complex configurations" by "for the most complex conditions".

Also perhaps reformulate the sentence to make it clearer for the reader (it's a bit long).

Response: We agree that keeping the wording consistent will allow for an unambiguous interpretation of our testing apparatus. We have adjusted the wording accordingly, using "configuration" when discussing the horizontal nature of the test and "condition" when referring to the single-string-rewarded nature of the test (L37, L261-262, L292-293).

We have also reformulated this sentence by cutting it in half to improve its readability (L37-42).

Reviewer 1: Line 42: provide more than one example in brackets

Response: In addition to the Werdenich and Huber (2006) reference we had originally included on keas spontaneously solving string-pull tests, we have added two other references showing the same behaviours in New Caledonian crows (Taylor et al. 2010) and in siskins and goldfinches (Seibt and Wickler 2006) on L44.

Reviewer 1: Line 49: correct: Charadriiformes not Charadriformes

Response: We thank the reviewer for picking up on this error and we have adjusted the spelling of Charadriiformes accordingly (L49).

Reviewer 1: Line 131: content of Figure 2 does not appear correctly in the manuscript

Response: We thank the reviewer for bringing up this concern. When Figure 2 is uploaded as a stand-alone PDF during the submission/proofing process, it appears to come out as intended, which is not the case when uploading it from the word file.

Reviewer 1: Line 251-252: try to stay consistent in throughout the manuscript, either use number or text e.g. you wrote "4 of them" but above you use text for numbers

Response: We thank the reviewer for picking up on this error. We now spelled out all numbers below 10, unless they are a measurement (time, dimension, mass) associated with a unit. Changes can be found on L158, L199, L225, L237, L239, L243, L248, L249, L250, L269, L272.

Reviewer 1: Line 263: please be more specific, see my comment above, you should specify horizontal configuration and only one “basic” condition (often not used during testing, but during training).

Response: We have clarified that the string-pull test used here was “the horizontal configuration, single-rewarded string condition of the string-pull test” on L261-262.

Reviewer 1: Line 288-289: perhaps do not speak about means-end understanding now, just say the importance of administering further conditions to properly assess the mechanisms at play

Response: This is a good point. We have removed the means-end understanding wording on L288 and we have added a sentence on L290-291 explaining that more complex conditions of the string-pull paradigm are required to understand the mechanisms that might underlie wild gulls’ probability of solving these tests.

Reviewer 1: Line 292-293: reformulate, see my comment above

Response: We agree and we have reformulated the sentence to specify that our subjects solved the horizontal configuration of the string-pull test using a single-baited string (L292-293).

Reviewer 1: Line 341: “We suggest that various configurations of the string-pull test outlined here should be administered”?

Response: We have changed this sentence accordingly to “...various configurations and conditions of the string-pull test outlined here...” on L340-341.

Reviewer 1: Line 344: remove “avian and”

Response: We have removed “avian and” accordingly on L344.

XX

We thank you for accepting our manuscript and hope that you find these minor revisions acceptable.

Sincerely,

On behalf of Jessika Lamarre and David Wilson